# Comparative Transcriptome Investigation of *Nosema ceranae* Infecting Eastern Honey Bee Workers

**DOI:** 10.3390/insects13030241

**Published:** 2022-02-28

**Authors:** Yuanchan Fan, Jie Wang, Kejun Yu, Wende Zhang, Zongbing Cai, Minghui Sun, Ying Hu, Xiao Zhao, Cuiling Xiong, Qingsheng Niu, Dafu Chen, Rui Guo

**Affiliations:** 1College of Animal Sciences (College of Bee Science), Fujian Agriculture and Forestry University, Fuzhou 350002, China; fanyc19980201@126.com (Y.F.); wanglegejie@163.com (J.W.); kejunyu@m.fafu.edu.cn (K.Y.); wdebee@163.com (W.Z.); caizongbing@126.com (Z.C.); hnmhsun@126.com (M.S.); hy19971020@126.com (Y.H.); f81967518@163.com (X.Z.); pandaxiongcl@sina.com (C.X.); 2Jilin Province Institute of Apicultural Science, Jilin 132000, China; apis1969@163.com; 3Apitherapy Research Institute, Fujian Agriculture and Forestry University, Fuzhou 350002, China

**Keywords:** honey bee, *Apis cerana cerana*, *Nosema ceranae*, microsporidian, transcriptome, differentially expressed gene, infection mechanism

## Abstract

**Simple Summary:**

At present, interaction between *Nosema ceranae* and *Apis cerana* is poorly understood, though *A. cerana* is the original host for *N. ceranae*. Here, comparative investigation was conducted using transcriptome data from *N. ceranae* infecting *Apis cerana cerana* workers at seven days post inoculation (dpi) and 10 dpi (NcT1 and NcT2 groups) as well as *N. ceranae* spores (NcCK group). There were 1411, 604, and 38 DEGs identified in NcCK vs. NcT1, NcCK vs. NcT2, and NcT1 vs. NcT2 comparison groups. Additionally, 10 upregulated genes and nine downregulated ones were shared by above-mentioned comparison groups. GO classification and KEGG pathway analysis suggested that these DEGs were engaged in a number of key functional terms and pathways such as cell part and glycolysis. Further analysis indicated that most of virulence factor-encoding genes were upregulated, while a few were downregulated during the fungal infection. Findings in this current work provide a basis for clarifying the molecular mechanism udnerlying *N. ceranae* infection and host-microsporidian interaction during bee nosemosis.

**Abstract:**

*Apis cerana* is the original host for *Nosema ceranae*, a widespread fungal parasite resulting in honey bee nosemosis, which leads to severe losses to the apiculture industry throughout the world. However, knowledge of *N. ceranae* infecting eastern honey bees is extremely limited. Currently, the mechanism underlying *N. ceranae* infection is still largely unknown. Based on our previously gained high-quality transcriptome datasets derived from *N. ceranae* spores (NcCK group), *N. ceranae* infecting *Apis cerana cerana* workers at seven days post inoculation (dpi) and 10 dpi (NcT1 and NcT2 groups), comparative transcriptomic investigation was conducted in this work, with a focus on virulence factor-associated differentially expressed genes (DEGs). Microscopic observation showed that the midguts of *A. c. cerana* workers were effectively infected after inoculation with clean spores of *N. ceranae*. In total, 1411, 604, and 38 DEGs were identified from NcCK vs. NcT1, NcCK vs. NcT2, and NcT1 vs. NcT2 comparison groups. Venn analysis showed that 10 upregulated genes and nine downregulated ones were shared by the aforementioned comparison groups. The GO category indicated that these DEGs were involved in a series of functional terms relevant to biological process, cellular component, and molecular function such as metabolic process, cell part, and catalytic activity. Additionally, KEGG pathway analysis suggested that the DEGs were engaged in an array of pathways of great importance such as metabolic pathway, glycolysis, and the biosynthesis of secondary metabolites. Furthermore, expression clustering analysis demonstrated that the majority of genes encoding virulence factors such as ricin B lectins and polar tube proteins displayed apparent upregulation, whereas a few virulence factor-associated genes such as hexokinase gene and 6-phosphofructokinase gene presented downregulation during the fungal infection. Finally, the expression trend of 14 DEGs was confirmed by RT-qPCR, validating the reliability of our transcriptome datasets. These results together demonstrated that an overall alteration of the transcriptome of *N. ceranae* occurred during the infection of *A. c. cerana* workers, and most of the virulence factor-related genes were induced to activation to promote the fungal invasion. Our findings not only lay a foundation for clarifying the molecular mechanism underlying *N. ceranae* infection of eastern honey bee workers and microsporidian–host interaction.

## 1. Introduction

*Nosema ceranae* is an obligate unicellular fungal parasite that specifically infects honey bee midgut epithelial cells. *N. ceranae* was first identified in eastern honey bee (*Apis cerana*) by Fries et al. [1], thereafter it swiftly spread to western honey bee (*Apis mellifera*) colonies reared in Europe and Taiwan province, China [2]. Currently, *N. ceranae* could be detected in colonies all over the world. *N. ceranae* infestation results in a battery of negative impacts on the honey bee host such as shortened life span, energy stress, immunosuppression, cell apoptosis inhibition [3,4,5,6], earlier foraging activity, and impaired navigation and cognitive ability [7,8]. A close connection between *N. ceranae* and colony collapse disorder (CCD) has been suggested by several studies [9,10].

*N. ceranae* exists outside the host cell only as dormant spores. After ingestion by the honey bee host, the inside polar tube is rapidly extruded to pierce the cell membrane followed by injection of the infective sporoplasm via hollow polar tubes [1,11,12]. The intracellular life cycle of *N. ceranae* can be divided into two phases including the proliferative phase (merogony) and the sporogonic phase (sporogony), and ends with the formation of spores. Nevertheless, it is hard to completely isolate *N. ceranae* at the above-mentioned two different phases, which is a key factor limiting further study on the fungal parasite during the infection process. Advances in next-generation sequencing technology have allowed for a deeper understanding of host response and parasite/pathogen infection. Previously, several studies have been conducted to investigate responses of western honey bee workers to microsporidian infestation, where Badaoui et al. analyzed the gene expression of *A. m. ligustica* workers at 5, 10, and 15 days post *N. ceranae* infection by RNA-Seq technology, and found that the expression of genes encoding host antimicrobial peptides, cuticle proteins, and odor binding proteins were downregulated, resulting in the decline in immune function of the host to promote the survival and propagation of *N. ceranae* at the colony level [13]; based on fluorescence in situ hybridization (FISH) and immunostaining experiments, Panek et al. investigated the impact of *N. ceranae* on *A. m. ligustica* workers’ epithelium renewal by following the mitotic index of midgut stem cells during a 22-day *N. ceranae* infection, and their results showed that *N. ceranae* can negatively alter the gut epithelium renewal rate and disrupt some signaling pathways involved in the gut homeostasis [14]. Transcriptome analysis of the intestinal tract of *A. c. cerana* exposed to *N. ceranae* demonstrated that microsporidian infection inhibited genes relevant to homeostasis and renewal in the wnt signaling pathway [15]. Comparatively, omics study on the *N. ceranae* infecting honey bee hosts is very limited, and the underlying mechanism of *N. ceranae* infection is still vague.

In our previous work, we performed RNA sequencing of clean spores of *N. ceranae*, followed by transcriptome-wide identification of ncRNAs such as miRNAs, lncRNAs, and circRNAs in *N. ceranae* spores [16,17,18]. Recently, we conducted deep sequencing of the midguts of *A. c. cerana* workers at 7- and 10-days post inoculation (dpi) with *N. ceranae* spores, and deciphered the host cellular and humoral responses to fungal infection [19]; in addition, we analyzed the expression profile of highly expressed genes (HEGs) and discussed their potential roles in *N. ceranae* infestation [20]. In light of the fact that transcriptomic study on *N. ceranae* during the infection process is still lagging behind, in the present study, based on the obtained transcriptome datasets, we first filtered out the parasite-derived data and then performed a comparative investigation combined with transcriptome data derived from *N. ceranae* spores, followed by deep investigation of the dynamics of genes in *N. ceranae* as well as the virulence factor-associated pathways and genes. To the best of our knowledge, this is the first report of omics study on *N. ceranae* invading eastern honey bees.

## 2. Materials and Methods

### 2.1. Fungal Spore and Honey Bee

Clean spores of *N. ceranae* were previously purified using Percoll discontinuous gradient centrifugation protocol and preserved in the Honey Bee Protection Laboratory [18], College of Animal Sciences (College of Bee Science). *A. c. cerana* workers were selected from three colonies located in the teaching apiary of the College of Animal Sciences (College of Bee Science) in Fujian Agriculture and Forestry University. No Varroa mite was observed during the whole study. The disappearance of seven common honey bee viruses (KBV, IAPV, ABPV, DWV, SBV, BQCV, and CBPV) and two honey bee microsporidia (*Nosema apis* and *N. ceranae*) in the newly emergent *A. c. cerana* workers was verified with the RT-PCR assay [21].

### 2.2. Microscopic Observation and PCR Validation of N. ceranae Spores

The prepared spores of *N. ceranae* were subjected to microscopic observation using an optical microscope (SIOM, Shanghai, China). Furthermore, the total RNA of spores were isolated and used as templates for reverse transcription; the resulting cDNA was then used as templates for PCR amplification with previously described specific primers for *N. ceranae* and *N. apis* [22,23] and the amplified products were detected by 1.5% agarose gel electrophoresis (AGE). Sterile water was set as a negative control.

### 2.3. Preparation and Detection of Paraffin Section of Honey Bee Midgut Tissue

One-day-old workers of *A. c. cerana* in *N. ceranae*-inoculated groups were each artificially fed with 5 μL 50% (*w*/*v*) sucrose solution containing 1 × 10^6^ spores; while 1-old-day workers in un-inoculated groups were each fed with 5 μL of 50% (*w*/*v*) sucrose solution without spores. At 11 dpi with *N. ceranae* spores, the midgut tissues in the *N. ceranae*-inoculated and un-inoculated groups were respectively harvested and fixed with 4% paraformaldehyde. According to our previously described protocol [21], on the basis of a microtome (Leica, Nussloch, Germany) and an embedding center (Junjie, Wuhan, China), paraffin sections were prepared and then stained with a hematoxylin eosin (HE) stain by Shanghai Sangon Biological Engineering Co. Ltd., followed by detection utilizing an optical microscope with a digital camera (SOPTOP, Shanghai, China).

### 2.4. Transcriptome Data Source

Midgut tissues of *A. c. cerana* workers at 7 dpi and 10 dpi with *N. ceranae* spores and corresponding un-inoculated worker’s midgut tissues was previously prepared and sequenced on an Illumina HiSeq^TM^ 4000 platform with a stand-specific cDNA library-based strategy [17]. The raw datasets were deposited in the National Biotechnology Information Center (NCBI) SRA database (https://www.ncbi.nlm.nih.gov/sra) (accessed on 21 July 2018) under BioProject number: PRJNA395264. Quality control of the produced raw reads were previously conducted according to the method described by Chen et al. [17,24]. Briefly, raw reads containing adapters, more than 10% of unknown nucleotides (N), and more than 50% of low quality (*q* value ≤20) bases were removed to gain high-quality clean reads, which were then mapped to the ribosome RNA (rRNA) database (www.arb-silva.de/) (accessed on 20 March 2021) using Bowtie2 software [24]. The result indicated that 174, 700, 032; 205, 297, 946; 124, 216, 829 and 99, 030, 788 raw reads were generated, and after quality control, 171, 868, 061; 200, 570, 776; 121, 949, 977 and 97, 432, 267 clean reads were gained, with Q20 of 94.96%, 94.58%, 94.60%, and 94.91%, respectively [19] (Appendix A). Hence, the high-quality RNA-Seq data can be used for transcriptomic analyses in the present study. The clean reads were first mapped to the reference genome of *Apis cerana* (assembly ACSNU-2.0) to remove host-derived data, and the unmapped clean reads were further mapped to the *N. ceranae* reference genome (assembly ASM98816v1) to gain microsporidian-derived data.

In another work, we conducted deep sequencing of clean spores of *N. ceranae* (NcCK) using Illumina HiSeq-based RNA-Seq [18]. A total of 416, 156, and 600 raw reads were produced, and after quality control, 210, 824, and 312 clean reads were obtained, with average a Q30 of 92.63%, which suggested that the high-quality transcriptome data can be used in this study. The clean reads were mapped to the reference genome of *N. ceranae* (assembly ASM98816v1). The relevant raw data were deposited in the NCBI SRA database (https://www.ncbi.nlm.nih.gov/sra) (accessed on 29 Augest 2019) under BioProject number: PRJNA562784 (Appendix A).

### 2.5. Identification and Analysis of DEGs in N. ceranae

Following the standard of *p* value ≤ 0.05 and |log_2_(Fold change)| ≥ 1, the DEGs in NcCK vs. NcT1 and NcCK vs. NcT2 comparison groups were screened by edgeR software [25]. Venn analysis of up- and downregulated genes in these comparison groups and expression cluster analysis were carried out based on OmicShare platform (https://www.omicshare.com/) (accessed on 10 April 2021). GO (Gene Ontology) categorization of DEGs was carried out using WEGO software [26]. Blastall tool was employed to conduct pathway analysis by comparing DEGs against the KEGG (Kyoto Encyclopedia of Genes and Genomes) database (https://www.kegg.jp/) (accessed on 10 April 2021) [27].

### 2.6. Investigation of Virulence Factor-Associated DEGs

On the basis of associated documentations with *N. ceranae* and findings from our previous studies on *N. ceranae* [16,17,18,19,20,23,28,29,30], virulence factors such as spore wall protein, ricin B lectin, chitinase, polar tube protein, phosphofructokinase, ATP/ADP translocase, ABC transporter, hexokinase, and pyruvate kinase and associated DEGs were selected for further investigation. Expression clustering analysis of aforementioned DEGs was performed utilizing the OmicShare platform.

### 2.7. RT-qPCR Validation of DEGs

Fifteen DEGs were randomly selected from NcCK vs. NcT1 (XM_002994928.1, XM_002996348.1, XM_002996299.1, XM_002996655.1, XM_002996468.1 and XM_002996253.1), NcCK vs. NcT2 (XM_002996303.1, XM_002996294.1, XM_002995682.1, XM_002996538.1, XM_002996655.1 and XM_002996253.1), and NcT1 vs. NcT2 (XM_002996294.1 and XM_002996468.1) comparison groups and subjected to RT-qPCR validation. The *actin* gene (gene6001) in *N. ceranae* was used as an internal reference. Specific forward and reverse primers for these DEGs and *actin* were designed with primer premier 5 (Appendix A). Total RNA of *N. ceranae* spores and the *N. ceranae*-inoculated workers’ midguts at 7 dpi and 10 dpi were respectively isolated using the RNA Extraction Kit (TaKaRa Company, Dalian, China). cDNA was synthesized through reverse transcription with the oligo dT primer and used as templates for the qPCR assay, which was carried out on a QuanStudio RealTime PCR System (ThemoFisher, Walthem, MA, USA). The qPCR reaction was conducted according to the instructions of the SYBR Green Dye Kit (Vazyme Company, Shanghai, China). Cycling parameters were as follows: 95 °C for 1 min, followed by 40 cycles at 95 °C for 15 s, 55 °C for 30 s, and 72 °C for 45 s. The relative gene expression was calculated based on the 2^−ΔΔCt^ method [31]. The experiment was performed three times utilizing three independent biological samples.

### 2.8. Statistical Analysis

Statistical analyses were conducted with SPSS software (IBM, Armonk, NY, USA) and GraphPad Prism 7.0 software (GraphPad, San Diego, CA, USA). Data were presented as mean ± standard deviation (SD). Statistical analysis was performed using the Student’s *t*-test and one-way ANOVA. Additionally, significant (*p* < 0.05) GO terms and KEGG pathways were filtered by performing Fisher’s exact test with R software 3.3.1.

## 3. Results

### 3.1. Verification of Infection of A. c. cerana Worker by N. ceranae

Under an optical microscope, oval and highly refractive dispersed spores were observed (Figure 1A). Furthermore, AGE indicated that the expected fragment (approximately 76 bp) was amplified from the purified spores with specific primers for *N. ceranae*, while no signal band was detected using specific primers for *N. apis* (Figure 1B). These results verified that the purified spores were indeed *N. ceranae* spores.

Based on microscopic observation of paraffin sections, it was found that there were a number of *N. ceranae* spores in the *A. c. cerana* worker’s midgut epithelial cells at 11 dpi with *N. ceranae* (Figure 2A,B), whereas no fungal spores could be detected in the worker’s midgut epithelial cells at 11 dpi without *N. ceranae* (Figure 2C,D). In addition, the structure of the midgut epithelial cells of *N. ceranae*-infected workers was fragmentary and the nucleic acid substances were dispersed and unclear (Figure 2A,B), while that of the un-infected worker was intact and the deeply colored cell nucleus was visible (Figure 2C,D). The results together suggest that the *A. c. cerana* workers were infected by *N. ceranae* and the host midgut epithelial cell structure was destroyed by the fungal infection.

### 3.2. Differential Gene Expression Profile of N. ceranae Infecting A. c. cerana Workers

In total, 1411, 604, and 38 DEGs were identified in the NcCK vs. NcT1, NcCK vs. NcT2, and NcT1 vs. NcT2 comparison groups, respectively. The numbers of upregulated genes were 711, 240, and 17, while those of the downregulated ones were 700, 360, and 21, respectively (Figure 3A). Additionally, Venn analysis showed that there were 10 and nine shared up- and downregulated genes in the aforementioned three comparison groups, whereas the numbers of unique upregulated (downregulated) genes were 417 (354), two (20), and five (10), respectively (Figure 3B). It is speculated that these shared DEGs play a common role during the *N. ceranae* infection process, whereas the unique DEGs exert specific function at different time points of *N. ceranae* invasion. Moreover, the expression clustering analysis showed that expression levels of ten shared DEGs such as ricin b lectin encoding genes (XM_002996294.1 and XM_002995387.1) were continuously upregulated, while another nine shared DEGs were continuously downregulated during the N. ceranae infection, indicating that some shared genes were activated by the *N. ceranae* infection while other shared genes were inhibited (Figure 3C).

### 3.3. Function and Pathway Annotation of DEGs in N. ceranae Infesting A. c. cerana Workers

GO term analysis suggested that the DEGs in NcCK vs. NcT1 comparison group were engaged in 638 biological process-associated functional terms such as metabolic process, single-organism process, and cellular process; 377 cellular component-associated terms such as cell, cell part, and organelle; and 424 molecular function-associated terms such as catalytic activity, transport, and binding (Figure 4A). In the NcCK vs. NcT2 comparison group, the DEGs could annotate to 22 GO terms including 291 biological process-related items such as metabolic process and single-organism, 160 cellular component-related terms such as cell and cell part, and 237 molecular function-related items such as catalytic activity and nucleic acid binding transcription factor activity (Figure 4B). The DEGs in the NcT1 vs. NcT2 comparison group were involved in seven functional terms such as metabolic process, cell, and catalytic activity (Figure 4C).

In addition, the KEGG pathway analysis indicated that the DEGs in the NcCK vs. NcT1 comparison group were relevant to 241 pathways, among these the most abundant groups were metabolic pathway, biosynthesis of secondary metabolites, ribosome, ribosome biogenesis in eukaryotes, and biosynthesis of antibiotics (Figure 5A). In the NcCK vs. NcT2 comparison group, the DEGs were relative to 201 pathways including metabolic pathway, biosynthesis of secondary metabolites, ribosome, ribosome biogenesis in eukaryotes, and biosynthesis of antibiotics (Figure 5B). The DEGs in the NcT1 vs. NcT2 comparison group were associated with 32 pathways such as metabolic pathway, biosynthesis of antibiotics, biosynthesis of secondary metabolites, glycolysis/gluconeogenesis, and microbial metabolism in diverse environments (Figure 5C).

### 3.4. Virulence Factor-Associated DEGs in N. ceranae Invading A. c. cerana Workers

Further investigation was conducted to explore virulence factor-associated DEGs in above-mentioned comparison groups, where a total of 20 DEGs were identified including six spore wall protein coding genes, three ricin B lectin protein coding genes, three ATP/ADP translocase protein coding genes, two polar tube protein coding genes, two ABC transporter protein coding genes, one chitin synthase protein coding gene, one 6-phosphofructokinase protein coding genes, one hexokinase protein coding gene, and one pyruvate kinase protein coding gene. Moreover, the expression clustering showed that the majority of virulence factor-encoding genes were induced to activation during the infection process, whereas a few were suppressed to a large extent (Figure 6).

### 3.5. Verification of DEGs via RT-qPCR

Fifteen DEGs were randomly selected for RT-qPCR validation, where the results suggest that the expression trend of 14 was consistent with those in transcriptome data (Figure 7A,B), confirming the reliability of the sequencing data used in this current work.

## 4. Discussion

Here, in the NcCK vs. NcT1 and NcCK vs. NcT2 comparison groups, 1411 and 604 DEGs were respectively identified including 657 and 240 upregulated genes as well as 700 and 364 downregulated ones. This is indicative of the overall alteration of genes in *N. ceranae* during the infection process. The response of fungal cells to external signals is regulated by mitogen-activated protein kinase (MAPK), which phosphorylates many downstream proteins and further alters gene expression, followed by participation in various processes such as proliferation, differentiation, and apoptosis [32]. Here, three and two DEGs in NcCK vs. NcT1 and NcCK vs. NcT2 were found to enrich in the MAPK signaling pathway, among which XM_002995908.1 (log_2_FC = 4.44, 4.93) and XM_002996683.1 (log_2_FC = 4.63, 4.51) were shared by both comparison groups, while there was a unique DEG (XM_002996282.1, log_2_FC = −1.01) in NcCK vs. NcT1, indicating that the MAPK signaling pathway was significantly activated during *N. ceranae* infection and used by the fungal parasite to respond to the internal environment of host epithelial cells. Previously, we conducted transcriptomic investigation of the *A. m. ligustica* workers’ midguts responding to *N. ceranae* invasion, and detected that seven (GenBank accession no. 9423510, 424610, 9424462, 9422987, 9422306, 9411598, and 9424007) DEGs in the midguts at 7 dpi and 10 dpi were enriched in the MAPK signaling pathway [30]. These results together demonstrate that the MAPK signaling pathway is likely to play a vital part in *N. ceranae* infection, but there were differences in this signaling pathway in *N. ceranae* when infecting different honey bee species. Ricin, a type of toxic heterodimer protein in castor seed, has a ricin chain B (RTB) that connects a ricin chain A (RTA) via a disulfide bond [33]. Lectins are proteins that recognize the glycan ligand. As virulence factors of many pathogens, lectins exert function by mediating cell adhesion, innate immune defense, and pathogen infection [34]. The binding of ricin B lectin (RBL) contributes to microbial infection and promotes pathogen attachment or entry into host cells. NBRBL3, a lectin protein of *Nosema bombyx*, was confirmed to be secreted from the spore during fungal proliferation and then mediate host cell recognition and adhesion [34]. In this research, three ricin B lectin-associated DEGs were observed in both the NcCK vs. NcT1 and NcCK vs. NcT2 comparison groups including XM_002995387.1 (log_2_FC = 4.52, 5.67), XM_002996294.1 (log_2_FC = 4.66, 5.88), and XM_002996299.1 (log_2_FC = 2.31, 2.96), and all of these were apparently upregulated, implying that *N. ceranae* may synthesize and secrete ricin B lectin to increase the adhesion between spores and host cells, further promoting the fungal proliferation [30].

There is a hard spore wall outside the microsporidia, which is composed of the outer spore wall layer, inner spore wall layer, and fibrous plasma membrane. Spore wall protein (SWP) has been proven to be a crucial virulence factor of microsporidia, which can recognize and adhere to host cells during microsporidian infection [35]. The spore wall protein SWP25 of *N. bombyx* was located in the endospore layer with a signal peptide, which was suggested to be implicated in chitin interaction of endospore powder and spore wall construction via the heparin binding motif (HBM) [36]. Here, one SWP25-encoding gene (XM_002996293.1) showed an upregulation trend in host midguts at 7 dpi (log_2_FC = 4.82) and 10 dpi (log_2_FC = 4.77), suggestive of the participation of SWP25 in *N. ceranae* infection of *A. c. cerana* workers. Yang et al. analyzed the interaction between *N**. bombycis* SWP9 (NbSWP9) with polar tube proteins PTP1 and PTP2 and found that NbSWP9 was mainly distributed in the polar tubes [37]. In this work, one gene encoding SWP9 (XM_002996348.1) was downregulated in the workers’ midguts at both 7 dpi (log_2_FC = −3.04) and 10 dpi (log_2_FC = −3.40), indicating that *A. c. cerana* workers may inhibit the interaction between SWP9 with PTPs via host–microsporidian interaction. However, additional work is required to decipher the underlying mechanism. Additionally, another two SWP encoding genes (XM_002996303.1, log_2_FC = 7.45, 7.90; XM_002994928.1, log_2_FC =5.08, 5.82), one SWP precursor encoding gene (XM_002996552.1, log_2_FC = 6.36, 6.90), and one spore wall and anchoring disk complex protein encoding gene (XM_002995858.1, log_2_FC = 3.41, 5.80) were identified and all were observed to upregulate in host midguts at both 7 dpi and 10 dpi, which demonstrated that these SWP-associated genes were activated to increase the synthesis of corresponding SWPs, further enhancing the *N. ceranae* invasion. For microsporidia, the infective protoplasm is transferred from the spore to the host cell through the polar tube, further starting the proliferation process, thus the polar tube protein is also considered as a virulence factor of importance. Yang et al. discovered that the polar tube and spore wall of microsporidia are the main components of mature spores adhering to and infecting the host cells [38]. Long et al. performed lectin blotting and β-elimination reaction of NbPTP1 in *N. bombycis*, where the results showed that NbPTP1 had O-glycosylation modification characteristics, which are conducive to adhesion, infection, and maintenance of the stability of the polar tube [39]. Here, two PTP1 encoding genes shared by the aforementioned two comparison groups were found to be upregulated including XM_002995447.1 (log_2_FC = 6.01, 6.84) and XM_002995446.1 (log_2_FC = 5.71, 6.48), indicating that both genes were induced to activation during the infection process to promote the proliferation of *N. ceranae*.

Due to the specific life cycle, microsporidia lost the typical mitochondria during long-term evolution, which was replaced by mitosis, an organelle that produces a small amount of ATP via glycolysis to meet the basic physiological needs [11]. Accordingly, microsporidia highly depend on stealing energy from the host cells, and glycolysis is the main manner for *N. ceranae* to produce ATP at a pretty low efficiency [12]. Huang et al. previously detected that the genes encoding three rate-limiting enzymes involved in catalyzing the glycolysis pathway continued to upregulate including hexokinase, pyruvate kinase, and phosphofructokinase [40]. Hexokinase catalyzes glucose phosphorylation to form glucose-6-phosphate (g-6-p), and the expression of hexokinase is regulated by ADP. Pyruvate kinase can catalyze the conversion of phosphoenolpyruvate to pyruvate and further enhance the production of ATP. Fructose-6-phosphate (f-6-p) is hydrolyzed by 6-phosphofructokinase, glucose-1 (F-1), and 6-diphosphate (6-2p) was inhibited by ATP with a high concentration [41,42,43]. Noticeably, the pyruvate kinase encoding gene XM_002996468.1 (log_2_FC = 2.81, 3.92) and hexokinase encoding gene XM_002995838.1 (log_2_FC = 3.78, 3.23) were upregulated in NcCK vs. NcT1 and NcCK vs. NcT2, which suggest that the activation of pyruvate kinase and hexokinase is a strategy of *N. ceranae* during the infection of *A. c cerana* workers. However, the 6-phosphofructokinase encoding gene XM_002994874.1 (log_2_FC = −4.72, −4.03) was downregulated, which was inconsistent with a previous study [40]. We inferred that it might be suppressed by ATP with high concentration. The underlying mechanism needs to be further explored.

ABC transporters represent the largest family of transmembrane proteins that contribute to ATP transportation. Most ABC transporters rely on ATP binding and hydrolysis to transport amino acids, lipids, sugars, peptides, ions, and other substrates from the cell fluid to the intracellular or extracellular region [44]. Here, the expression trend of two ABC transporter-encoding genes was altered in the above-mentioned two comparison groups: XM_002996245.1 (log_2_FC = 2.52, 2.86) was upregulated, while XM_002996253.1 (log_2_FC = −3.49, −3.68) was downregulated. This indicates that ABC transporters may play an essential role in the transmembrane transport of materials and energy during the *N. ceranae* infection, and different ABC transporters encoding genes may have different roles in this process. ATP/ADP translocase transports ATP synthesized in the mitochondrial matrix to cytoplasm for cell utilization in eukaryotes. *N. ceranae* needs to steal ATP from host cells by ATP/ADP translocase to meet its growth and development needs. Here, the upregulation of two ATP/ADP translocase genes was detected in NcCK vs. NcT1 and NcCK vs. NcT2 including XM_002995682.1 (log_2_FC = 3.82, 3.33) and XM_002996655.1 (log_2_FC = 3.70, 4.87), whereas another gene XM_002996538.1 (log_2_FC = −2.17, −2.08) was downregulated, indicative of the complex interaction between the microsporidian and host. The result supported the vital function of ATP/ADP transporters in the energy acquisition of *N. ceranae* invading *A. c. cerana* workers.

## 5. Conclusions

In summary, our results demonstrated that the overall transcriptome dynamics of *N. ceranae* was altered during the infection of *A. c. cerana* workers, and a number of virulence factor-associated genes were induced to activation to facilitate the fungal proliferation, but some other genes encoding virulence factors were suppressed via host–microsporidian interaction. Findings in this current work provide a basis for deciphering the mechanism underlying the *N. ceranae* infection of *A. c cerana* workers and microsporidian–host interaction.

## Figures and Tables

**Figure 1 insects-13-00241-f001:**
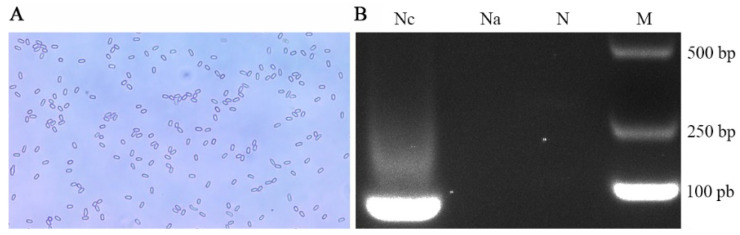
Microscopic detection and PCR validation of *N. ceranae* spores. (**A**) Microscopic detection (400 times amplification). (**B**) AGE for PCR amplified fragments, Lane Nc: Specific primers for *N. ceranae*, Lane Na: Specific primers for *N. apis*, Lane N: Sterile water (Negative control), Lane M: DNA marker.

**Figure 2 insects-13-00241-f002:**
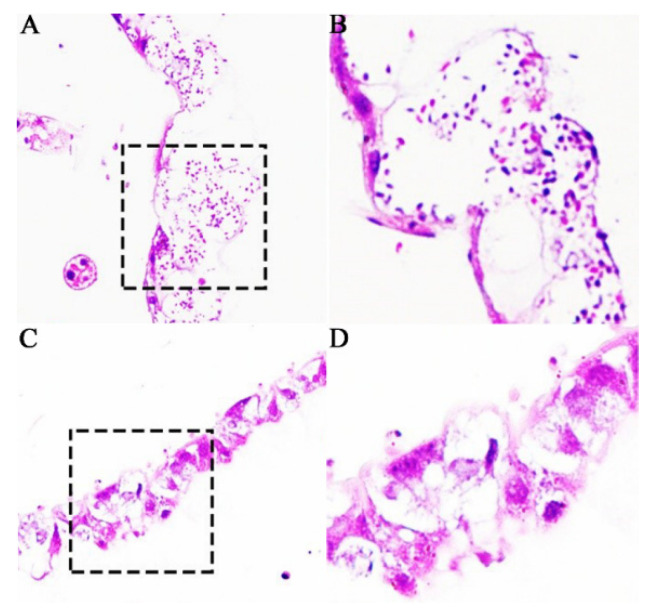
Microscopic observation of paraffin sections of *N. ceranae*-inoculated and un-inoculated *A. c. cerana* workers’ midguts. (**A**) Worker’s midgut at 11 dpi with *N. ceranae* under 200 times amplification. (**B**) Worker’s midgut at 11 dpi with *N. ceranae* under 400 times amplification. (**C**) Worker’s midgut at 11 dpi without *N. ceranae* under 200 times amplification. (**D**) Worker’s midgut at 11 dpi without *N. ceranae* under 400 times amplification. Black dashed box shows the region for observation under 400 times amplification.

**Figure 3 insects-13-00241-f003:**
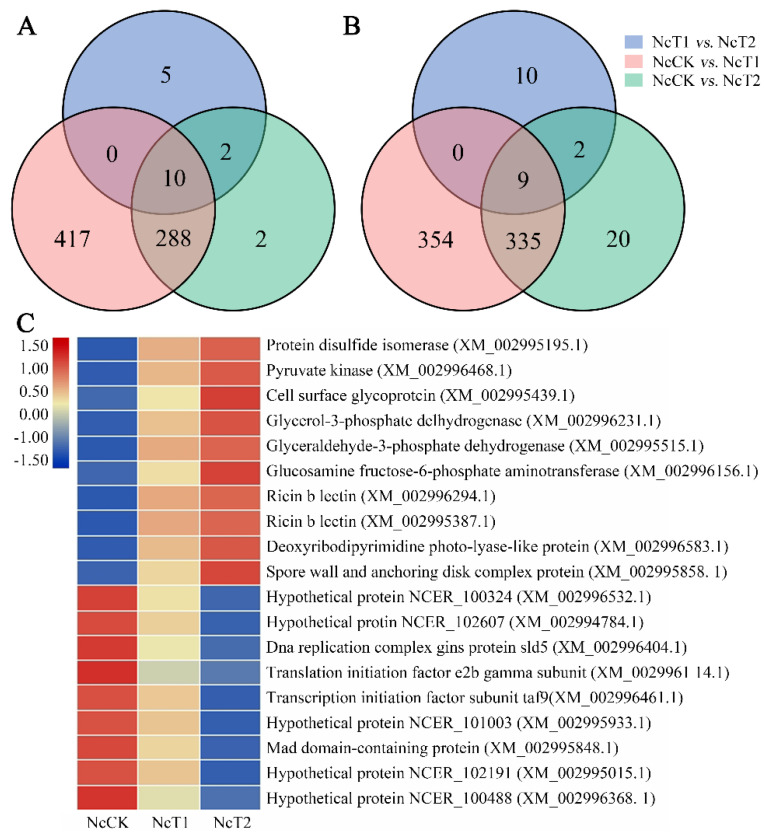
Analysis of DEGs in *N. ceranae.* (**A**) Venn analysis of upregulated genes in NcCK vs. NcT1, NcCK vs. NcT2, and NcT1 vs. NcT2 comparison groups. (**B**) Venn analysis of downregulated genes in NcCK vs. NcT1, NcCK vs. NcT2, and NcT1 vs. NcT2 comparison groups. (**C**) Heatmap of common DEGs in the NcCK, NcT1, and NcT2 groups.

**Figure 4 insects-13-00241-f004:**
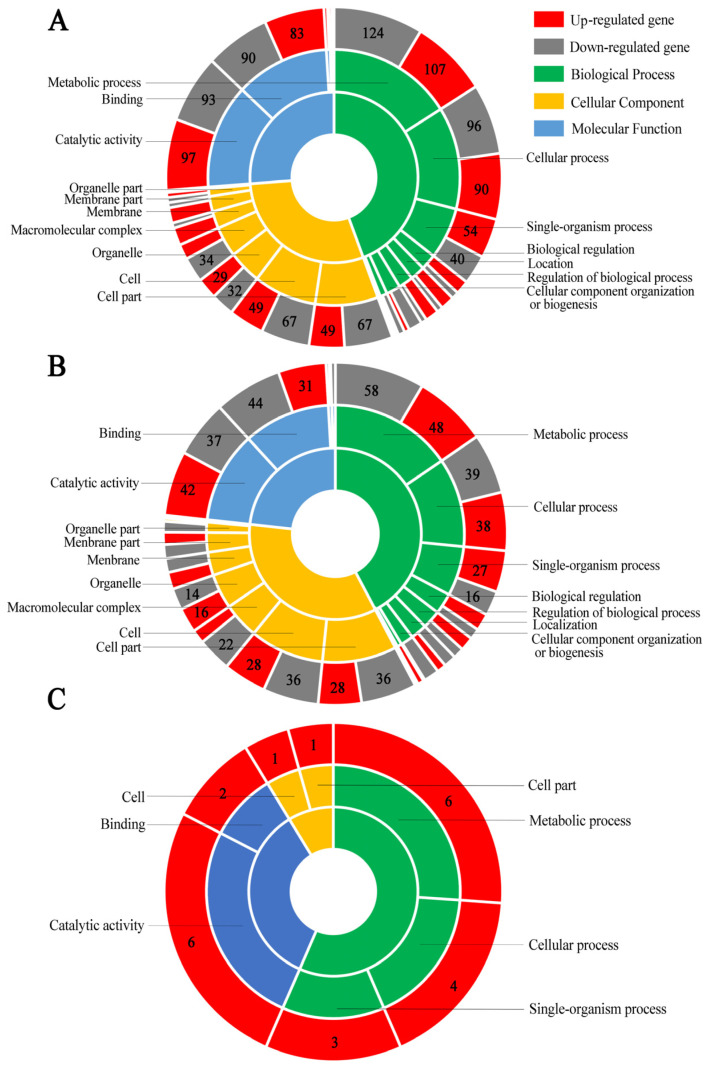
GO category of DEGs. (**A**) DEGs in NcCK vs. NcT1. (**B**) DEGs in NcCK vs. NcT2; (**C**) DEGs in NcT1 vs. NcT2.

**Figure 5 insects-13-00241-f005:**
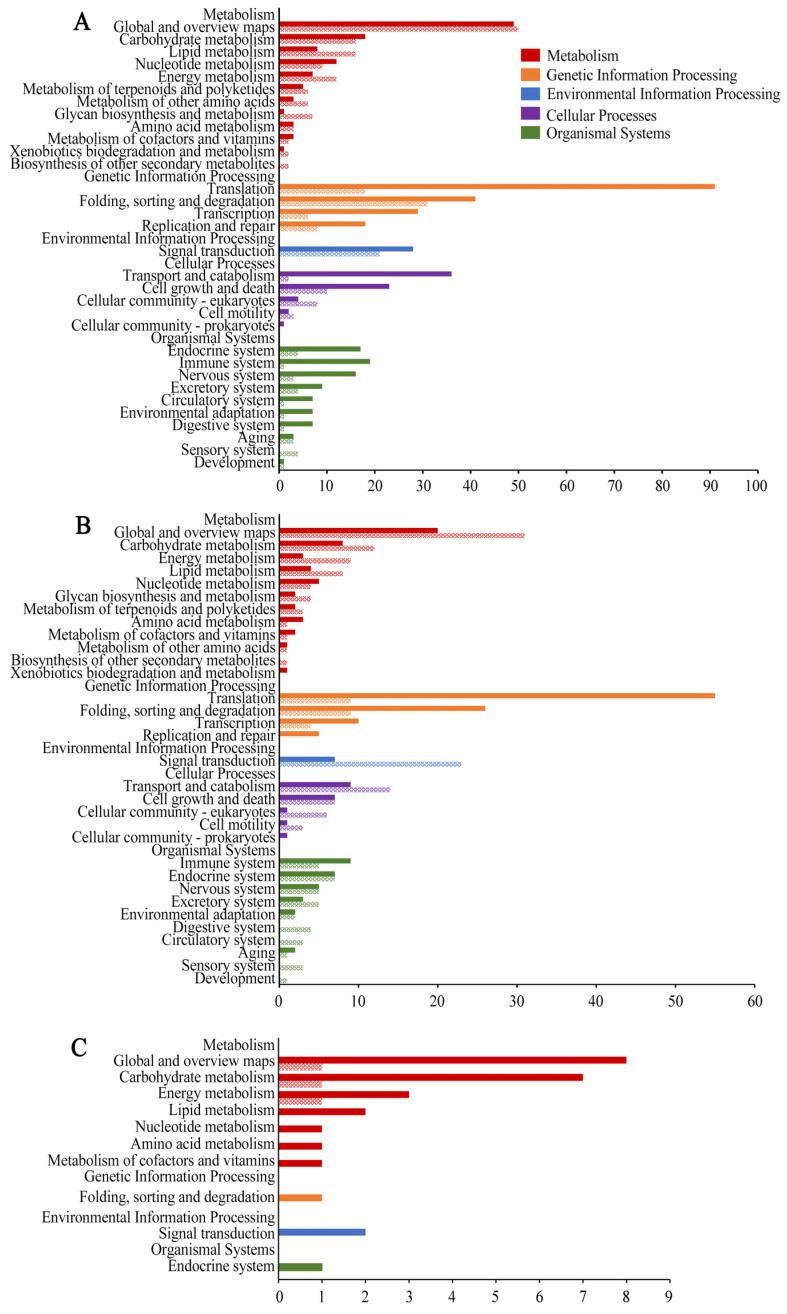
Pathways annotated by DEGs. (**A**) DEGs in NcCK vs. NcT1. (**B**) DEGs in NcCK vs. NcT2. (**C**) DEGs in NcT1 vs. NcT2.

**Figure 6 insects-13-00241-f006:**
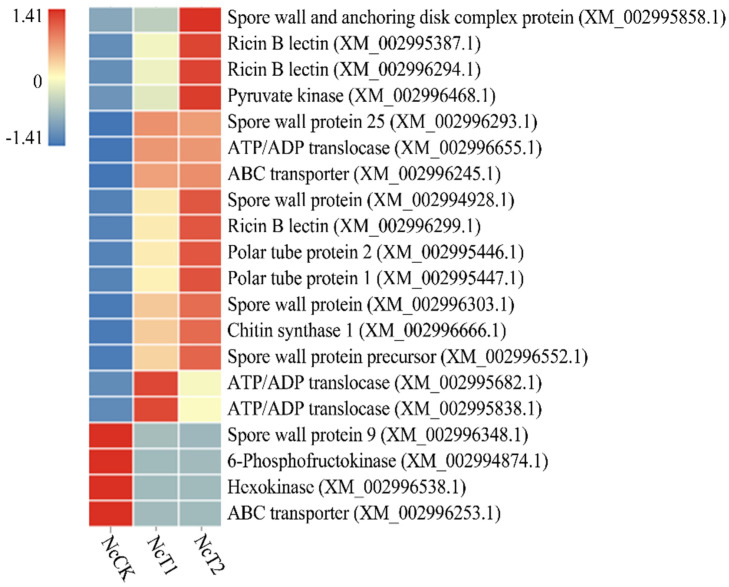
Heatmap of virulence factor-associated DEGs shared by the NcCK, NcT1, and NcT2 groups.

**Figure 7 insects-13-00241-f007:**
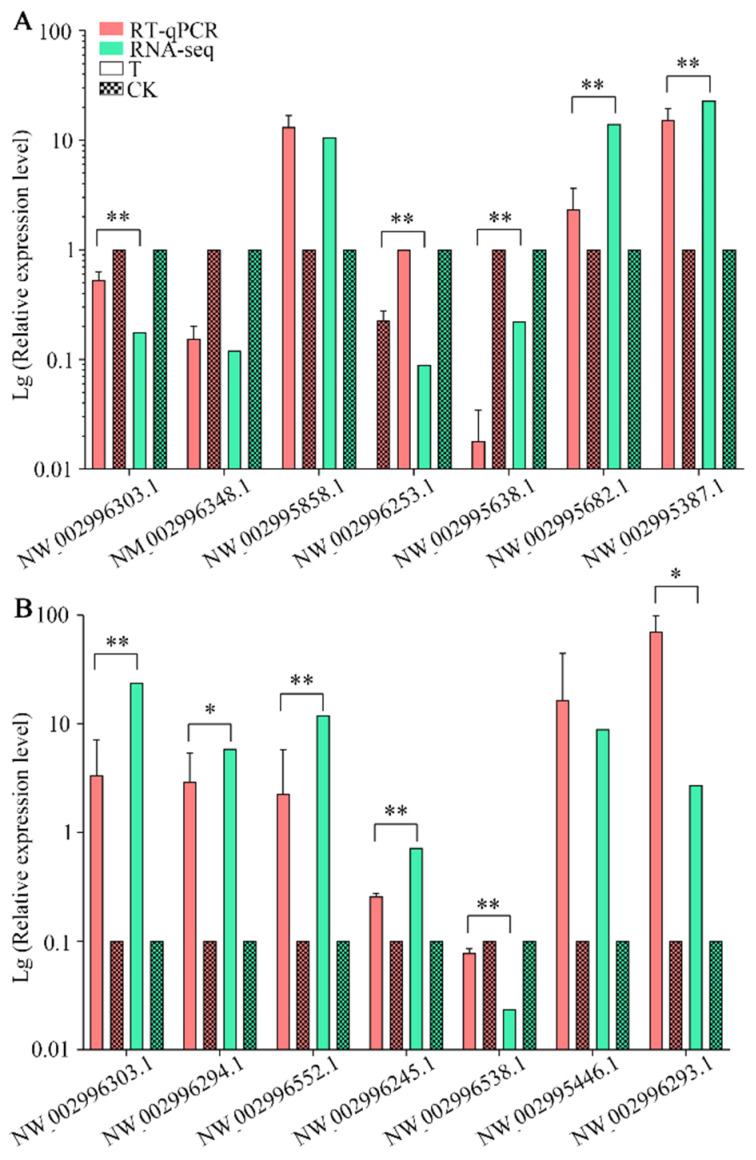
RT-qPCR verification of DEGs in *N. ceranae* infecting *A. c. cerana* workers. (**A**) DEGs in NcCK vs. NcT1. (**B**) DEGs in NcCK vs. NcT2. Error bars represent the variance of RT-qPCR results of each DEG. Bars with asterisk symbol indicate statistical differences (*p* < 0.05).

## Data Availability

The data presented in this study are available in article and Appendix A.

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
