# Peer review of "Comparative Transcriptome Investigation of Nosema ceranae Infecting Eastern Honey Bee Workers"

_insects, 2022, doi:10.3390/insects13030241_

Round 1

Reviewer 1 Report

Manuscript: insects-1538265 Comparative transcriptome investigation of Nosema ceranae infecting eastern honeybee workers

The authors performed comparative transcriptome investigation of Nosema ceranae infecting eastern honeybee workers. The authors demonstrated that the overall transcriptome dynamics of N. ceranae was altered during the infection of A. c. cerana workers, a number of virulence factor-associated genes were induced to activation to facilitate the fungal infestation, but some other genes encoding virulence factors were suppressed via host-microsporidian interaction. Totally, 1411, 604, and 38 DEGs were identified from NcCK vs. NcT1, NcCK vs. NcT2 and NcT1 vs. NcT2 comparison groups. Venn analysis showed that ten upregulated genes and nine down-regulated ones were shared by aforementioned comparison groups. GO category indicated these DEGs were involved in a series of functional terms relevant to biological process, cellular component, and molecular function, such as metabolic process, cell part, and catalytic activity. Additionally, KEGG pathway analysis suggested that the DEGs were engaged in an array of pathways of great importance, such as metabolic pathway, glycolysis, and biosynthesis of secondary metabolites. Further, expression clustering analysis demonstrated that majority of genes encoding virulence factors such as ricin B lectins and polar tube proteins displayed apparent up-regulation, whereas a few virulence factor-associated genes such as hexokinase gene and 6-phosphofructokinase gene presented down-regulation during the fungal infection. Finally, the expression trend of 14 DEGs was confirmed by RT-qPCR, validating the reliability of the transcriptome datasets. These results together demonstrated that an overall alteration of the transcriptome of N. ceranae occurred during the infection of A. c. ceranae workers, and most of virulence factor-related genes were induced to activation to promote the fungal invasion. Findings in this current work not only provide a basis for clarifying the mechanism underlying the N. ceranae infection of A. c cerana workers, but also shed light on the development of novel strategy for nosemosis control.

The data analysis methods are correct.
The English of the text is well written and well readable but needs additional checking with a professional translator.
The uniqueness of the text is more than 90% by AntiPlagiarism.NET.
The text contains some misspellings and typos.

There are some comments:
Line 2 - Comparative transcriptome investigation of Nosema ceranae infecting eastern honeybee workers >> Comparative transcriptome investigation of Nosema ceranae infecting eastern honeybee colonies.
Line 20 - Authors did not explain in detail what is NcT1, and NcT2 groups.
Line 35 - Authors said in Abstract - the results shed light on developing novel targets for microsporidiosis control. Could you make some detail how we can use novel targets for microsporidiosis control?

Line 41 - add the citation after cells >> cells (Ilyasov et al., 2012; Ilyasov et al., 2013). Add to the References following articles:
Ilyasov, R.A., Gaifullina, L.R., Saltykova, E.S., Poskryakov, A.V., Nikolenko, A.G., 2012. Review of the expression of antimicrobial peptide defensin in honey bees Apis mellifera L. Journal of Apicultural Science 56 (1), 115-124. doi: 10.2478/v10289-012-0013-y.
Ilyasov, R.A., Gaifullina, L.R., Saltykova, E.S., Poskryakov, A.V., Nikolaenko, A.G., 2013. Defensins in the honeybee antiinfectious protection. Journal of Evolutionary Biochemistry and Physiology 49 (1), 1-9. doi: 10.1134/S0022093013010015.

Line 47 - add the citation after ability >> ability (Ilyasov et al., 2012; Ilyasov et al., 2013). Add to the References following articles:
Ilyasov, R.A., Gaifullina, L.R., Saltykova, E.S., Poskryakov, A.V., Nikolenko, A.G., 2012. Review of the expression of antimicrobial peptide defensin in honey bees Apis mellifera L. Journal of Apicultural Science 56 (1), 115-124. doi: 10.2478/v10289-012-0013-y.
Ilyasov, R.A., Gaifullina, L.R., Saltykova, E.S., Poskryakov, A.V., Nikolaenko, A.G., 2013. Defensins in the honeybee antiinfectious protection. Journal of Evolutionary Biochemistry and Physiology 49 (1), 1-9. doi: 10.1134/S0022093013010015.

Line 97 - was >> were
Line 120 - were >> was
Line 141 - were >> was
Line 269 - delete our them
Line 277 - indictive >> is indicative
Line 284 - response >> respond
Line 300 - of >> in
Line 363 - Is the mitosis organelle name?
Line 363 - organell >> organelle
Line 385 - proteinsm >> proteins

Please improve the manuscript according to the above comments.
No other comments.

A minor revision is required.

Author Response

Dear Reviewers,

We appreciate your comments and suggestions of great importance, which significantly improve the quality of our work and manuscript. Accordingly, we seriously checked and modified the manuscript, and all revision were showed in red in the revised version of manuscript. Point-to-point response to review comments were as follows:

Point 1: Line 2 - Comparative transcriptome investigation of Nosema ceranae infecting eastern honeybee workers >> Comparative transcriptome investigation of Nosema ceranae infecting eastern honeybee colonies.

Response 1: Thanks for your kind suggestion. In fact, for the sake of avoiding interference from environmental factors, we previously conducted the inoculation of Apis cerana cerana workers with Nosema ceranae spores, rearing of workers, and sample preparation under lab conditions, not in reared colonies. Hence, we think the title “Comparative transcriptome investigation of Nosema ceranae infecting eastern honeybee workers” is more appropriate.

Point 2: Line 20 - Authors did not explain in detail what is NcT1, and NcT2 groups.

Response 2: Following your kind comment, we added related description to give detailed information for NcT1 group and NcT2 group.

Point 3: Line 35 - Authors said in Abstract - the results shed light on developing novel targets for microsporidiosis control. Could you make some detail how we can use novel targets for microsporidiosis control?

Response 3: Thank you for your valuable recommendation, based on which we improved the Abstract. Meanwhile, we enhanced the content regarding how use novel targets for microsporidiosis control.

Point 4: Line 41 - add the citation after cells >> cells (Ilyasov et al., 2012; Ilyasov et al., 2013). Add to the References following articles:

Ilyasov, R.A., Gaifullina, L.R., Saltykova, E.S., Poskryakov, A.V., Nikolenko, A.G., 2012. Review of the expression of antimicrobial peptide defensin in honey bees Apis mellifera L. Journal of Apicultural Science 56 (1), 115-124. doi: 10.2478/v10289-012-0013-y.

Ilyasov, R.A., Gaifullina, L.R., Saltykova, E.S., Poskryakov, A.V., Nikolaenko, A.G., 2013. Defensins in the honeybee antiinfectious protection. Journal of Evolutionary Biochemistry and Physiology 49 (1), 1-9. doi: 10.1134/S0022093013010015.

Response 4Thank you very much for your recommendations. We thought these references are not appropriate in this manuscript.

Point 5: Line 47 - add the citation after ability >> ability (Ilyasov et al., 2012; Ilyasov et al., 2013). Add to the References following articles:

Ilyasov, R.A., Gaifullina, L.R., Saltykova, E.S., Poskryakov, A.V., Nikolenko, A.G., 2012. Review of the expression of antimicrobial peptide defensin in honey bees Apis mellifera L. Journal of Apicultural Science 56 (1), 115-124. doi: 10.2478/v10289-012-0013-y.

Ilyasov, R.A., Gaifullina, L.R., Saltykova, E.S., Poskryakov, A.V., Nikolaenko, A.G., 2013. Defensins in the honeybee antiinfectious protection. Journal of Evolutionary Biochemistry and Physiology 49 (1), 1-9. doi: 10.1134/S0022093013010015.

Response 5: Thanks.  We thought these references are not appropriate in this manuscript.

Point 6: Line 97 - was >> were

Response 6: Corresponding correction was made following your comment.

Point 7: Line 120 - were >> was

Response 7: Corresponding modification was made according to your comment.

Point 8: Line 141 - were >> was

Response 8: We made the correction in the revised manuscript.

Point 9: Line 269 - delete our them

Response 9: We made the deletion following your kind advice.

Point 10: Line 277 - indictive >> is indicative

Response 10: We made the correction in the revised manuscript.

Point 11: Line 284 - response >> respond

Response 11: We replaced “response” with “respond” following your comment.

Point 12: Line 300 - of >> in

Response 12: We made corresponding correction in the revised manuscript.

Point 13: Line 363 - organell >> organelle

Response 13: Corresponding modification was made according to your kind comment.

Point 14: Line 385 - proteinsm >> proteins

Response 14: Corresponding correction was performed following your kind comment.

Point 15: Line 363 - Is the mitosis organelle name?

Response 14: The “mitosis” mentioned here is a mistake, and it should be replaced by word “mitosome”, which is an organelle specific for microsporidia.

Reviewer 2 Report

This manuscript discussed a perspective of Nosema ceranae transcriptome during its infection of Eastern honey bees. This topic is somehow interesting in this field. However, there some points have to clarify and quality of writing should be improved before publication.  

Line 19-20 what are NcCK, NcT1, NcCK, NcT2, NcT1 and NcT2?

Line 31: N. ceranae à Italic

Line 125-127: The numbers of reads are hard to distinguish, please summary the sequencing data in a table, including the raw reads, QC reads, mappability etc.

Line 114: section 2.4: which reference genome of the RNA-seq data were mapped?

Line 137-138 what are NcT1, NcT2?

Line 159: …actin were designed with primer premier 5 (Table SI)…is that honey bee actin? Or microsporidia actin? Besides the target gene names should added to the primer information of Table SI.

Line 167: …The relative gene expression was calculated based on 2–ΔΔCt method… how to calculate the  2–ΔΔCt of mature spore?

Figure 1 and 2: please add the scale bar in Figure 1A.

Figure 3: The meaning of the comparison is not clear please explain it and add this information in the results.

Figure 3B and 6: I don’t understand the basic calculation of the heatmap and the unit of the heatmap, please label it. If it was based on the DEG, then the labels should be NcCK-T1, NcCK-T2 and T1-T2?

Figure 5: in this figure, there are two different bar styles, please   

Figure 7: Some of the validated genes were not the same in two group (NcCK-T1 and NcCK-T2), what is the selection criteria? And also the presenting of RNA-seq data were based on Figure 3B or Figure 6? It is difficult to match the data.

Figure 7: Each data point should be presented in the bar to clear display the distribution of variation.

Author Response

Dear Reviewers,

We appreciate your comments and suggestions of great importance, which significantly improve the quality of our work and manuscript. Accordingly, we seriously checked and modified the manuscript, and all revision were showed in red in the revised version of manuscript. Point-to-point response to review comments were as follows:

Point 1: Line 19-20 what are NcCK, NcT1, NcCK, NcT2, NcT1 and NcT2?

Response 1: Thanks for your kind comment. In fact, NcCK group indicates purified spores of Nosema ceranae; whereas NcT1 and NcT2 groups indicate Nosema ceranae infecting the midguts of Apis cerana cerana workers at 7 days post inoculation (dpi) and 10 dpi with Nosema ceranae spores. Necessary explanation was added into the Abstract to provide detailed information for NcCK, NcT1, and NcT2 groups.

Point 2: Line 31: N. ceranae à Italic.

Response 2: Following your kind comment, we made thorough check throughout the whole manuscript and necessary correction.

Point 3: Line 125-127: The numbers of reads are hard to distinguish, please summary the sequencing data in a table, including the raw reads, QC reads, mappability etc.

Response 3: In view of that the result of transcriptome data quality control was presented in Table format in our previously published paper (19.21. Fu, Z.; Zhou, D.; Chen, H.; Geng, S.; Chen, D.; Zhen, Y.; Xiong, C.; Xu, G.; Zhang, X.; Guo, R. Analysis of highly expressed genes in Apis cerana cerana workers midguts responding to Nosema ceranae stress. Journal of Sichuan University (Natural Science Edition) 2020, 57, 191-198, doi: 10.3969/j.issn.0490-6756.2020.01.029.ï¼›17.19. Chen, D.; Chen, H.; Du, Y.; Zhou, D.; Geng, S.; Wang, H.; Wan, J.; Xiong, C.; Zheng, Y.; Guo, R. Genome-wide identification of long non-coding RNAs and their regulatory networks involved in Apis mellifera ligustica response to Nosema ceranae infection. Insects 2019, 10, 245, doi: 10.3390/insects10080245), we just summarized the result in the Materials and Methods section in this manuscript to give necessary information. Thanks.

Point 4: Line 114: section 2.4: which reference genome of the RNA-seq data were mapped?

Response 4: The transcriptome data in this study was mapped to the reference genome of Nosema ceranae (assembly ASM18298v1). Corresponding information was added in section 2.4.

Point 5: Line 159: …actin were designed with primer premier 5 (Table SI)…is that honey bee actin? Or microsporidia actin? Besides the target gene names should added to the primer information of Table SI.

Response 5: The internal reference gene actin mentioned here was Nosema ceranae actin. Based on your kind recommendation, we added the gene names to the primer information described in Table SI. Thanks.

Point 6: line 167: …The relative gene expression was calculated based on 2–ΔΔCt method… how to calculate the 2–ΔΔCt of mature spore?

Response 6: As described in Materials and Methods section 2.7, Total RNA of N. ceranae spores and N. ceranae-inoculated workers’ midguts at 7 dpi and 10 dpi were respectively isolated followed by cDNA synthesis and qPCR. Six DEGs were randomly selected from NcCK vs. NcT1, while another six DEGs were randomly selected from NcCK vs. NcT2 comparison group. The relative expression levels of above mentioned DEGs were calculated using 2–ΔΔCt method by comparing the expression of DEGs in N. ceranae-inoculated workers’ midguts with those in N. ceranae spores. Thanks for your kind comment.  

Point 7: Figure 1 and 2: please add the scale bar in Figure 1A.

Response 7: When performing microscopic observation of purified spores and paraffin sections, we photographed using software but forget to add the scale bar at the meantime. In the legends of Figure 1 and Figure 2, we indicated the amplification times for microscopic detection. During our study in the future, we will remember to add the scale bar when performing microscopic observation. Thanks for your recommendation of importance.

Point 8: Figure 3: The meaning of the comparison is not clear please explain it and add this information in the results.

Response 8: On basis of your valuable comment, we seriously thought and explore the deep meaning. Please see the description associated with Figure 3 in the revised manuscript.

Point 9: Figure 3B and 6: I don’t understand the basic calculation of the heatmap and the unit of the heatmap, please label it. If it was based on the DEG, then the labels should be NcCK-T1, NcCK-T2 and T1-T2?

Response 9: There were 1411, 604, and 38 DEGs in NcCK vs. NcT1, NcCK vs. NcT2, and NcT1 vs. NcT2 comparison groups. In view of the numbers of DEGs in each comparison group, we didn’t conduct expression clustering of total DEGs in each comparison group, instead we performed expression clustering of those DEGs (totally 19) shared by three comparison groups. Hence the labels below the heatmap is correct. For a heatmap, different colors represent the alteration trend of genes, thus there is no unit. In Figure 3C, the upper 10 genes displayed a continuous up-regulation trend, while the lower nine genes showed a continuous down-regulation trend. On the basis of Figure 3C, we can detect a clear alteration trend of 19 expressed genes shared by NcCK, NcT1, and NcT2 groups.

Point 10: Figure 5: in this figure, there are two different bar styles, please

Response 10: Following your significant advice, we improved the legend to give more clear information.

Point 11: Figure 7: Some of the validated genes were not the same in two group (NcCK-T1 and NcCK-T2), what is the selection criteria? And also the presenting of RNA-seq data were based on Figure 3B or Figure 6? It is difficult to match the data. Figure 7: Each data point should be presented in the bar to clear display the distribution of variation.

Response 11: The genes we screened for RT-qPCR validation were randomly screened based on differential gene expression levels and were not associated with Figure 3B and Figure 6. The purpose is to verify the accuracy of the data. Due to the large differences in the expression levels of different genes, each data point cannot be displayed completely, and there is no correlation between the genes we screened, so we use a truncated histogram to display the data, which can accurately display the data.

Round 2

Reviewer 2 Report

Figure 1 and 2: can you take a new picture with a scale bar to represent the size of the mature spore? Or added the detail of what fold (200x or 400x?) of the observation under microscopy.

Author Response

Figure 1 and 2: can you take a new picture with a scale bar to represent the size of the mature spore? Or added the detail of what fold (200x or 400x?) of the observation under microscopy.

Response: Thanks for your kind recommendation, following which we added necessary information about amplification times into the legend of Figure 1 and Figure 2.